# Evaluation of primary health care by users during the COVID-19 pandemic: A cross-sectional study

**Suely Deysny de Matos Celino**[1]*, **Nailton José Brandão de Albuquerque Filho**[2], **Monalisa da Nóbrega Cesarino Gomes**[3], **Gabriela Maria Cavalcanti Costa**[4], **Ana Elza Oliveira de Mendonça**[1]

**1** Graduate Program in Collective Health, UFRN, Natal, Rio Grande do Norte, Brazil, **2** Department of Physical Education, State University of Paraíba, Campina Grande, Paraíba, Brazil, **3** Department of Dentistry, Centro Universitário Unifacisa, Campina Grande, Paraíba, Brazil, **4** Public Health Program, State University of Paraíba, Campina Grande, Paraíba, Brazil

* deysny@hotmail.com

## Abstract

**Data Availability Statement:** All relevant data are within the paper.

### Objective

To evaluate the primary health care (PHC) attributes and associated factors during the COVID-19 pandemic using the perspective of users.

### Methods

This cross-sectional, quantitative study included 422 PHC users from 96 Family Health Teams in a city in Brazil. The assessment used the Primary Care Assessment Tool (PCA-Tool) and a structured questionnaire on the sociodemographic and epidemiological characteristics of users and basic health units (BHU). The Person's chi-square test was used to analyze the association between high overall scores in PCATool and characteristics of users and BHU. Crude and adjusted prevalence ratios (PR) with a 95% confidence interval were also calculated. Poisson regression and Rao Scott's Chi-square test were used to estimate crude PR.

### Results

Most users were aged 30 to 39 years (26.3%), women (75.4%), registered at the BHU for over ten years (59.5%), and had incomplete secondary education (30.6%). The mean of PHC essential attributes and overall scores were low (6.10 ± 0.81 and 5.78 ± 0.77, respectively). "First-contact care–use" received the highest score (9.22 ± 1.62), while "first-contact care–accessibility" received the lowest (2.82 ± 0.90). High overall scores were associated with an average employment time of professionals (doctors and nurses) at the BHU (PR = 1.31; 95% CI 1.17–1.48; p < 0.001) and lower educational level of users (PR = 1.71; 95% CI 1.54–1.90; p < 0.001).

**Funding:** This study was partly financed by the Coordenacão de Aperfeiçoamento de Pessoal de Nıvel Superior– Brasil(CAPES)– Finance Code 001. The funding consisted of a payment of publication fees. Furthermore, it did not interfere with the study's design and collection, analysis, data interpretation, and manuscript writing. There is no additional external funding received for this study.

**Competing interests:** The authors declare no conflicts of interest.

## Conclusion

"First-contact care–use" was the best evaluated, while "first-contact care–accessibility" was the worst. High scores were associated with a lower educational level of users and BHU with more experienced professionals.

## Introduction

In December 2019, Wuhan, the capital of Hubei, China, reported the first coronavirus disease 2019 (COVID-19) cases. The disease presented quick spread and severity, reaching other province regions and worldwide, characterizing a pandemic after China, Germany, Japan, Vietnam, and the United States registered human-to-human transmission [1]. Then, in March 2020, the World Health Organization (WHO) declared the global outbreak of COVID-19 as an international emergency and recommended the implementation of consistent and evidence-based decisions [2].

Although differently in each country, the COVID-19 impacted health services worldwide, since they were unprepared for a pandemic [3]. In Brazil, the Ministry of Health published ordinances, manuals, guides, and protocols with information on symptoms, transmission forms, clinical management, and guidelines to prevent COVID-19 infection [4]. In April 2020, the "protocol for clinical management of coronavirus (COVID-19) in primary health care (PHC)" was published and aimed to "define the role of PHC services in the management and control of COVID-19 infection, and to provide clinical guidance tools for professionals" but without instructions for restructuring actions of PHC [5]. The PHC, by being first contact of the population with the health system, was essential during the COVID-19 pandemic and its reorganization was specially challenging [6].

Thus, assessing the pandemic impact on the quality of care and work process of PHC teams is important to improve action plans and adaptation. According to Starfield [7], a PHC assessment requires identifying whether PHC attributes guided the health services, improving health indicators, user satisfaction, costs, equity, and the health status of the population.

The Primary Care Assessment Tool (PCATool) is a structured questionnaire developed to assess PHC essential (first-contact care, longitudinality, comprehensiveness, and coordination) and derived (family and community orientation and cultural competence) attributes using the perspective of users and health professionals. The instrument is in the public domain, used by WHO, and was validated in different countries (e.g., Brazil, South Korea, and Spain) [8].

The perception of the quality of health services is determined by sociodemographic characteristics of users and inherent factors [9]. The federal government has been seeking strategies to strengthen PHC in Brazil, such as the "Mais Médicos" program and the promotion of family medicine and primary care residencies. These approaches help the training, provision, and settlement of medical doctors in remote areas while ensuring the effectiveness of care using incentives for professional training [10].

In 2019, the Ministry of Health instituted the "*Saúde na Hora*" program [11], which implemented extended hours in health units to increase access to health services. Additionally, Murray and Berwick [12] proposed Advanced Access, a strategy to improve user satisfaction within PHC. The slogan was 'Do today the work of today!', allowing users to meet their health needs, preferably on the same day, avoiding fragmentation of care [13].

Despite studies discussing PHC assessment using the PCATool [14–16], applying it in a pandemic is relevant to identify weaknesses and potentialities during public health emergencies. Moreover, associating the satisfaction of users with sociodemographic and service

characteristics may reveal important aspects of public health policies. Thus, we hypothesized that worse sociodemographic factors and PHC conditions would be associated with unmet attributes. The study aimed to assess PHC during the COVID-19 pandemic, considering its attributes and using the perspective of users and associated factors.

## Materials and methods

This cross-sectional, quantitative study was conducted between August 2021 and June 2022 in 96 family health teams (FHT) from Campina Grande, Paraíba, Brazil, which had 334,187 users [17]. Sample size calculation was performed using the Survey System platform (available at https://www.surveysystem.com/sscalc.htm) and adopting the following criteria: 95% confidence interval (95% CI), 5% error, and 50% estimated prevalence. The sample size was set at 348 users. Considering possible losses and inadequate questionnaire completion, 10% was added to the sample size estimation. Thus, 422 users were divided into seven sanitary districts of Campina Grande.

Data collection began in 2021 when the municipality had 110 FHT (comprising a medical doctor, a nurse, a nursing technician, and community health agents) distributed in 86 basic health units (BHU) and seven health districts, covering 91.78% of the Family Health Strategy [17].

Despite the wide territorial coverage, formal guidelines, contingency plans, or municipal decrees of measures for PHC were not established in Campina Grande. The municipality restructured the hospital network, establishing an emergency care unit as a reference for COVID-19 screening and the municipal hospital for admissions, guiding PHC teams to refer for those services [18].

Inclusion criteria were users over 18 years old, registered in an FHT for at least six months, vaccinated against COVID-19, and available for in-person data collection at the BHU. Due to the high contamination, the municipal health department demanded the vaccination of the researcher and users for in-person data collection. Users were selected proportionally to the number of registered users in the district and each BHU. Those with cognitive difficulties in answering the questionnaire.

Data collection was conducted by a single researcher and occurred at the BHU during consultation. A structured questionnaire created by the authors was used with questions about sociodemographic and epidemiological characteristics of users (sex, age, educational level, chronic disease, time registered at the BHU, COVID-19 infection, tested, monitored, and vaccinated for COVID-19 at the BHU). Questions related to the BHU were also included (type of community, partnership with the "*Mais Médicos*" or the "*Saúde na Hora*" programs, Family Medicine or Primary Care Residency, number of FHT, use of advanced access, average employment time of professionals [doctors and nurses] at the BHU, and COVID-19 testing).

A pilot study with 20 users from two BHU was conducted to verify the data collection methodology and dynamics; changes were unnecessary. Users from the pilot study were not included in the main study.

For the quantitative assessment of PHC, the expanded version for adults and the Brazilian population of the validated instrument Primary Care Assessment Tool (PCATool) was used [19]. PCATool provides an overall score and scores for the four PHC essential attributes (i.e., first-contact care, longitudinality, comprehensiveness, and coordination) and three derived attributes (i.e., family and community orientation and cultural competence). It evaluates the structure and process of care regarding each PHC attribute in a BHU using a Likert scale ("definitely yes," "probably yes," "probably not," "definitely not," and "I don't know/I don't remember"). Each score ranges from 1 to 4, which was transformed into values from 0 to 10 for data

analysis according to the formula [20]:

$$Essential\ or\ overall\ score = \frac{(obtained\ score - 1)}{(4 - 1)} \times 10$$

The essential score was calculated based on the average score of essential attributes, and the overall score was calculated based on the average score of all attributes. Scores were classified as high (score $\geq$ 6.6) or low (score $<$ 6.6), with higher scores indicating better PHC evaluation [20].

According to the Brazilian PCATool Manual [20], the percentage of items with missing values (blank values or answers with code "9"—"don't know, don't remember") should be under 50% of the total number of attribute items to calculate user score.

Absolute and relative frequencies were used to characterize users and BHU. PHC attributes were presented as mean, standard deviation, median, minimum, and maximum values. Pearson's Chi-square test assessed the association between PHC attributes with high overall scores and characteristics of users and BHU. Crude and adjusted prevalence ratios (PR) with 95% CI were estimated. Poisson regression and Rao Scott's chi-square test were used to estimate the crude PR. Poisson regression models with a robust estimator were used for the adjusted analysis of the outcome. All variables with a $p < 0.20$ in the crude PR analysis were included in the initial adjusted model, having been rounded to one decimal place; however, to remain in the final model, the variables presented $p < 0.05$. Analyzes were performed using Stata software, version 14.

The study was approved by the research ethics committee of the State University of Paraíba (Brazil) under process number 45288021.2.0000.5187 in compliance with Resolution 466/12 of the National Health Council. Users were informed of the study aims and signed the informed consent form; confidentiality and anonymity were guaranteed.

## Results

Table 1 shows the characteristics of the BHU and the 422 users. Most BHU were located in urban communities (87.7%). Regarding partnerships, most BHU were neither associated with the "*Mais Médicos*" (92.2%) nor with the "*Saúde na Hora*" programs (82.2%) and did not have Family Medicine or Primary Care Residency (69.7%). Most users were aged 30 to 39 years (26.3%), women (75.4%), had incomplete secondary education (30.6%), and were registered in the BHU for over ten years (59.5%) (Table 1).

Table 2 shows the mean, standard deviation, median, minimum, and maximum values of PHC attributes evaluated by users. The best-evaluated attribute was "first-contact care–use" (M = 9.22, SD = ± 1.62); half of the users gave a score of 10 to this attribute. The worst evaluated attribute was "first-contact care–accessibility" (M = 2.82, SD ± 0.90); half of the users gave a score equal to or less than 2.78 for this attribute. The average overall score was 5.78 (± 0.77); half of the users gave a score equal to or lower than 5.8 for the BHU (Table 2).

Table 3 shows the high ($\geq$ 6.6) and low ($<$ 6.6) overall scores distribution according to the characteristics of users and BHU. A higher frequency of high overall scores was observed in BHU without partnership with the "*Saúde na Hora*" program (p = 0.001), with Family Medicine or Primary Care Residency (p = 0.034), two FHTs (p = 0.044), and longer average employment time of professionals at the BHU (p < 0.001). Regarding characteristics of users, longer time registered at the BHU was the only characteristic associated with high overall scores (p = 0.025) (Table 3).

The adjusted PR analysis indicated that the factors associated with high overall scores were the average employment time of professionals at the BHU and the educational level of users. BHU

**Table 1. Characteristics of basic health units (BHU) and users.** Campina Grande, Paraiba, Brazil, 2022.

| Variables | n (%) |
|---|---|
| **BHU** | |
| *Type of community* | |
| Rural | 35 (8.3) |
| Urban | 370 (87.7) |
| Mixed | 17 (4.0) |
| *Partnership with the "Mais Médicos" program* | |
| Yes | 30 (7.1) |
| No | 392 (92.2) |
| *Partnership with the "Saúde na Hora" program* | |
| Yes | 75 (17.8) |
| No | 347 (82.2) |
| *Family Medicine or Primary Care Residency* | |
| Yes | 128 (30.3) |
| No | 294 (69.7) |
| *Number of family health teams* | |
| One | 212 (50.2) |
| Two | 167 (39.6) |
| Three | 43 (10.2) |
| Use of a*dvanced access* | |
| Yes | 105 (24.9) |
| No | 226 (53.6) |
| Partially | 91 (21.6) |
| *Average employment time of professionals (doctors and nurses) at the BHU* | |
| Less than one year | 70 (16.6) |
| Between one and five years | 158 (37.4) |
| Between five and ten years | 153 (36.3) |
| Over ten years | 41 (9.7) |
| *COVID testing* | |
| Yes | 86 (20.4) |
| No | 336 (79.6) |
| **Users** | |
| *Sex* | |
| Woman | 318 (75.4) |
| Man | 104 (24.6) |
| *Age group* | |
| 18 to 29 years | 95 (22.5) |
| 30 to 39 years | 111 (26.3) |
| 40 to 49 years | 93 (22.0) |
| 50 to 59 years | 62 (14.7) |
| 60 years or older | 61 (14.5) |
| *Educational level* | |
| Illiterate | 24 (5.7) |
| Incomplete secondary education | 129 (30.6) |
| Complete secondary education | 91 (21.6) |
| Incomplete high school | 79 (18.7) |
| Complete high school | 88 (20.9) |
| University education | 9 (2.1) |

*(Continued)*

**Table 1.** (Continued)

| Variables | n (%) |
|---|---|
| Postgraduate | 2 (0.5) |
| *Chronic disease* | |
| Hypertension | 93 (22.0) |
| Diabetes | 31 (7.3) |
| Mental disease | 91 (21.6) |
| Heart disease | 79 (18.7) |
| Obstructive lung disease | 88 (20.9) |
| Obesity | 9 (2.1) |
| No disease | 29 (6.87) |
| *Time registered at the BHU* | |
| Less than one year | 35 (8.3) |
| Between one and five years | 50 (11.8) |
| Between six and ten years | 86 (20.4) |
| Over ten years | 251 (59.5) |
| *Covid-19 infection* | |
| Yes | 119 (28.2) |
| No | 303 (71.8) |
| *Tested for COVID-19 at the BHU* | |
| Yes | 10 (8.5) |
| No | 109 (91.5) |
| *Monitored for COVID-19 by the BHU* | |
| Yes | 7 (5.9) |
| No | 112 (94.1) |
| *Vaccinated for COVID-19 at the BHU* | |
| Yes | 102 (24.2) |
| No | 320 (75.8) |

*PHC: primary health care; BHU: basic health unit

with professionals with an average employment time between one and five years showed a 12% higher prevalence of high overall scores (PR = 1.12, p = 0.021); between five to ten years showed a 21% higher prevalence of high overall scores (PR = 1.21, p < 0.001); and over ten years showed a 30% higher prevalence of high overall score (PR = 1.30, p < 0.001) than BHU with professionals with less than one year. Regarding the educational level of users, a high frequency of high overall scores was observed at lower educational levels. Illiterate users assigned high overall scores to the BHU, 70% more than postgraduate users (PR = 1.70, p < 0.001) (Table 4).

## Discussion

This study demonstrated that higher levels of education of users and longer employment time of professionals were associated with better PHC attributes. The municipality scored under the minimum score 6.6 for essential attributes, indicating services without the minimum PHC characteristics. A review study indicated a poor quality of PHC due to the COVID-19 pandemic, highlighting its damage to the quality of health services, especially due to reduced accessibility and increased health care interruption [21].

The attribute "first-contact care–use" obtained the best score, highlighting that users recognize the BHU as the first health care option. However, the attribute "first-contact care–

**Table 2. Mean, standard deviation, median, minimum, and maximum values of PHC attributes according to the users.** Campina Grande, Paraíba, Brazil, 2022.

| Attributes | Mean | Standard deviation | Median | Minimum | Maximum |
|---|---|---|---|---|---|
| Affiliation | 8.48 | 1.710 | 10.00 | 0.0 | 10.0 |
| First-contact care–use | 9.22 | 1.620 | 10.00 | 0.0 | 10.0 |
| First-contact care—accessibility | 2.82 | 0.903 | 2.78 | 0.6 | 5.5 |
| Longitudinality | 6.28 | 1.571 | 6.43 | 1.7 | 9.3 |
| Coordination—integrated care | 6.38 | 2.654 | 6.25 | 0.4 | 10.0 |
| Coordination—information system | 7.80 | 1.619 | 7.78 | 3.3 | 10.0 |
| Comprehensiveness–services available | 4.84 | 0.913 | 4.85 | 2.3 | 7.3 |
| Comprehensiveness—services provided | 3.51 | 1.268 | 3.33 | 0.0 | 7.2 |
| Family orientation | 5.67 | 1.930 | 5.56 | 0.0 | 10.0 |
| Community orientation | 3.44 | 1.178 | 3.33 | 0.0 | 7.8 |
| Essential score | 6.10 | 0.812 | 6.15 | 1.0 | 7.7 |
| Overall score | 5.78 | 0.768 | 5.81 | 2.2 | 7.6 |

*PHC: primary health care.

accessibility" obtained the worst score, similar to studies that investigated the impact of the COVID-19 pandemic on PHC in other countries [22, 23]. In Brazil, poor accessibility was present before the pandemic and reflected organizational issues (e.g., reduced opening hours, difficulty scheduling appointments, and long waits for care) [24].

A study conducted with PHC health professionals and managers of Brazilian municipal health departments observed actions on health surveillance and comprehensive care. The actions were performed via telephone, WhatsApp app, and home visits of community health agents to guarantee access and pandemic control by PHC. Moreover, the study indicated a barrier: need for more supply, internet, personal protective equipment, reverse transcription polymerase chain reaction (RT-PCR) tests, and continuing education for professionals [25].

Before the pandemic, the Ministry of Health implemented the "*Saúde na Hora*" program, aiming to expand the coverage of the Family Health Strategy by extending the working hours in BHU [11]. Some authors believe that extending the working hours increases accessibility, improving user satisfaction [24, 26]. However, the findings of our study indicated a higher frequency of high overall scores in services without partnership with the "*Saúde na Hora*" program, even during the pandemic. Moreover, the BHU associated with the program were the only ones with rapid tests for COVID-19, and their services focused on the demands of the emergency, impairing other health care actions.

"Coordination—information systems" was the second best-evaluated attribute by users. At the time of data collection, the municipality used the information system of the Ministry of Health for registration and monitoring of PHC users, which has the Electronic Record of Citizen, promoting effective coordination and management of care [27].

The average scores for "longitudinality" and "coordination—integrated care" attributes were slightly below the cutoff point. Most professionals had over a year of employment time, favoring bonding, which is fundamental to the longitudinality of care. However, poor accessibility to services and other attributes may hinder operationalization [28].

The attribute "coordination" improves the PHC quality by enhancing the accessibility to other levels of care, integrating actions, services, and professionals vertically and horizontally, and linking community and individual resources [29]. Poor care coordination hinders managing non-communicable and infectious diseases [30], especially during the COVID-19

**Table 3. High and low overall PCATool scores according to users and BHU characteristics.** Campina Grande, Paraíba, Brazil, 2022.

| Variables | High overall score | Low overall score | p-value[1] |
|---|---|---|---|
| | n (%) | n (%) | |
| **BHU** | | | |
| *Type of community* | | | |
| Rural | 16 (45.7) | 19 (54.3) | 0.849 |
| Urban | 186 (50.3) | 184 (49.7) | |
| Mixed | 9 (52.9) | 8 (47.1) | |
| *Partnership with the "Mais Médicos" program* | | | |
| Yes | 17 (56.7) | 13 (43.3) | 0.449 |
| No | 194 (49.5) | 198 (50.5) | |
| *Partnership with "Saúde na hora" program* | | | |
| Yes | 25 (33.3) | 50 (66.7) | 0.001 |
| No | 186 (53.6) | 161 (46.4) | |
| *Family Medicine or Primary Care Residency* | | | |
| Yes | 74 (57.8) | 54 (42.2) | 0.034 |
| No | 137 (46.6) | 157 (53.4) | |
| *Number of family health teams* | | | |
| One | 103 (48.6) | 109 (51.4) | 0.044 |
| Two | 93 (55.7) | 74 (44.3) | |
| Three | 15 (34.9) | 28 (65.1) | |
| *Use of advanced access* | | | |
| Yes | 57 (54.3) | 48 (45.7) | 0.008 |
| No | 98 (43.4) | 128 (56.6) | |
| Partially | 56 (61.5) | 35 (38.5) | |
| *Average employment time of professionals at the BHU* | | | |
| Less than one year | 21 (30.0) | 49 (70.0) | < 0.001 |
| Between one and five years | 72 (45.6) | 86 (54.4) | |
| Between five and ten years | 89 (58.2) | 64 (41.8) | |
| Over ten years | 26 (70.7) | 12 (29.3) | |
| *COVID testing* | | | |
| Yes | 38 (44.2) | 48 (55.8) | 0.227 |
| No | 173 (51,5) | 163 (48,5) | |
| **Users** | | | |
| *Sex* | | | |
| Woman | 48 (46.2) | 56 (53.8) | 0.366 |
| Man | 163 (51.3) | 155 (48.7) | |
| *Age group* | | | |
| 18 to 29 years | 49 (51.6) | 46 (48.4) | 0.217 |
| 30 to 39 years | 51 (45.9) | 60 (54.1) | |
| 40 to 49 years | 40 (43.0) | 53 (57.0) | |
| 50 to 59 years | 37 (59.7) | 25 (40.3) | |
| 60 years or older | 34 (55.7) | 27 (44.3) | |
| *Educational level* | | | |

(*Continued*)

**Table 3.** (Continued)

| Variables | High overall score | Low overall score | p-value[1] |
|---|---|---|---|
| | n (%) | n (%) | |
| Illiterate | 17 (70.8) | 7 (29.2) | 0.177 |
| Incomplete secondary education | 64 (49.6) | 65 (50.4) | |
| Complete secondary education | 50 (54.9) | 41 (45.1) | |
| Incomplete high school | 36 (45.6) | 43 (54.4) | |
| Complete high school | 39 (44.3) | 49 (55.7) | |
| University education | 5 (55.6) | 4 (44.4) | |
| Postgraduate | 0 | 2 (100.0) | |
| *Chronic disease* | | | |
| Hypertension | 46 (49.5) | 47 (50.5) | 0.317 |
| Diabetes | 21 (67.7) | 10 (32.3) | |
| Mental disease | 3 (33.3) | 6 (66.7) | |
| Heart disease | 3 (75.0) | 1 (25.0) | |
| Obstructive lung disease | 2 (33.3) | 4 (66.7) | |
| Obesity | 1 (33.3) | 2 (66.7) | |
| No disease | 135 (48.9) | 141 (51.1) | |
| *Time registered at the BHU* | | | |
| Less than one year | 13 (37.1) | 22 (62.9) | 0.025 |
| Between one and five years | 17 (34.0) | 33 (66.0) | |
| Between five and ten years | 47 (54.7) | 39 (45.3) | |
| Over ten years | 134 (53.4) | 117 (46.6) | |
| *COVID-19 infection* | 18 (64.3) | 10 (35.7) | |
| Yes | 94 (54.0) | 80 (46.0) | |
| No | | | |
| *Tested for COVID-19 at the BHU* | 60 (50.4) | 59 (49.6) | 0.913 |
| Yes | 139 (49.8) | 140 (50.2) | |
| No | | | |
| *Monitored for COVID-19 by the BHU* | 6 (60.0) | 4 (40.0) | 0.439 |
| Yes | 51 (47.2) | 57 (52.8) | |
| No | | | |
| *Vaccinated for COVID-19 at the BHU* | 4 (57.1) | 3 (42.9) | 0.646 |
| Yes | 53 (48.2) | 57 (51.8) | |
| No | | | |

* PCATool: Primary Care Assessment Tool; BHU: basic health unit;

[1]Pearson's chi-square test p < 0.05.

pandemic, when PHC is fundamental in organizing the care network and guaranteeing an efficient flow [31].

The reorganization of the PHC during the pandemic should aim at preserving its attributes. Thus, mild cases can be solved, and severe cases identified and referred to specialized care, maintaining care coordination [32]. In this study, PHC services were not considered the first level of contact of users with suspected or confirmed COVID-19 since our results indicated a lack of COVID testing, monitoring, and vaccination at the BHU, hindering an efficient line of care to guarantee care comprehensiveness.

A study conducted in Italy concluded that the rapid overcrowding of hospitals due to COVID-19 was caused by government negligence with PHC services and a lack of cooperation

**Table 4. Crude and adjusted PR using poisson regression for the association between high overall scores and characteristics of users and BHU.** Campina Grande, Paraíba, Brazil, 2022.

| Variables | Crude PR 95% CI | p-value | Adjusted PR 95% CI | p-value |
|---|---|---|---|---|
| **BHU** | | | | |
| *Type of community* | | | - | - |
| Rural | 1 | - | - | - |
| Urban | 1.03 (0.92–1.16) | 0.610 | - | - |
| Mixed | 1.05 (0.87–1.27) | 0.622 | | |
| *Partnership with the "Mais Médicos" program* | | | | |
| Yes | 1 | - | - | - |
| No | 0.95 (0.85–1.07) | 0.436 | - | - |
| *Partnership with "Saúde na hora" program* | | | | |
| Yes | 1 | - | - | - |
| No | 1.15 (1.06–1.26) | 0.001 | - | - |
| *Family Medicine or Primary Care Residency* | | | | |
| Yes | 1 | - | - | - |
| No | 0.93 (0.87–0.99) | 0.031 | - | - |
| *Number of family health teams* | | | | |
| One | 1 | - | - | - |
| Two | 1.05 (0.98–1.11) | 0.168 | - | - |
| Three | 0.91 (0.81–1.02) | 0.099 | - | - |
| *Use of advanced access* | | | | |
| Yes | 1 | - | - | - |
| No | 0.92 (0.86–1.00) | 0.060 | - | - |
| Partially | 1.05 (0.96–1.14) | 0.304 | - | - |
| *Average employment time of professionals (doctors and nurses) at the BHU* | | | | |
| Less than one year | 1 | - | 1 | - |
| Between one and five years | 1.12 (1.01–1.24) | 0.024 | 1.12 (1.02–1.24) | 0.021 |
| Between five and ten years | 1.22 (1.10–1.34) | < 0.001 | 1.21 (1.10–1.34) | < 0.001 |
| Over ten years | 1.31 (1–17–1.48) | < 0.001 | 1.30 (1.16–1.46) | < 0.001 |
| *COVID testing* | | | | |
| Yes | 1 | | - | - |
| No | 1.05 (0.97–1.14) | 0.232 | - | - |
| **Users** | | | | |
| *Sex* | | | | |
| Woman | 0.97 (0.90–1.04) | 0.370 | - | - |
| Man | 1 | - | - | - |
| *Age group* | | | | |
| 18 to 29 years | 1 | - | - | - |
| 30 to 39 years | 0.96 (0.88–1.06) | 0.419 | - | - |
| 40 to 49 years | 0.94 (0.86–1.04) | 0.239 | - | - |
| 50 to 59 years | 1.05 (0.95–1.17) | 0.314 | - | - |
| 60 years or older | 1.03 (0.93–1.14) | 0.610 | - | - |
| *Educational level* | | | | |
| Illiterate | 1.71 (1.54–1.90) | < 0.001 | 1.70 (1.52–1.91) | < 0.001 |
| Incomplete secondary education | 1.50 (1.41–1.59) | < 0.001 | 1.52 (1.40–1.65) | < 0.001 |
| Complete secondary education | 1.55 (1.45–1.66) | < 0.001 | 1.55 (1.40–1.65) | < 0.001 |
| Incomplete high school | 1.46 (1.35–1.57) | < 0.001 | 1.48 (1.35–1.62) | < 0.001 |
| Complete high school | 1.44 (1.34–1.55) | < 0.001 | 1.48 (1.35–1.62) | < 0.001 |

*(Continued)*

**Table 4.** (Continued)

| Variables | Crude PR 95% CI | p-value | Adjusted PR 95% CI | p-value |
|---|---|---|---|---|
| University education | 1.56 (1.26–1.91) | < 0.001 | 1.55 (1.23–1.96) | < 0.001 |
| Postgraduate | 1 | - | 1 | - |
| *Chronic disease* | | | | |
| Hypertension | 1.00 (0.93–1.09) | 0.927 | - | - |
| Diabetes | 1.13 (1.01–1.25) | 0.028 | - | - |
| Mental disease | 0.90 (0.71–1.13) | 0.356 | - | - |
| Heart disease | 1.18 (0.92–1.50) | 0.198 | - | - |
| Obstructive lung disease | 0.90 (0.67–1.19) | 0.449 | - | - |
| Obesity | 0.90 (0.60–1.33) | 0.590 | - | - |
| No disease | 1 | - | - | - |
| *Time registered at the BHU* | | | | |
| Less than one year | 1 | - | - | - |
| Between one and five years | 0.92 (0.51–1.63) | 0.765 | - | - |
| Between five and ten years | 1.47 (0.92–2.36) | 0.109 | - | - |
| Over ten years | 1.44 (0.92–2.25) | 0.111 | - | - |
| *COVID-19 infection* | | | | |
| Yes | 1 | - | - | - |
| No | 1.00 (0.98–1.02) | 0.978 | - | - |
| *Tested for COVID-19 at the BHU* | | | | |
| Yes | 1 | - | - | - |
| No | 0.94 (0.74–1.20) | 0.635 | - | - |
| *Monitored for COVID-19 by the BHU* | | | | |
| Yes | 1 | - | - | - |
| No | 0.97 (0.78–1.21) | 0.819 | - | - |

*PR: prevalence ratios; BHU: basic health units; BHU: basic health unit; p < 0.05

between this sector and other public providers and private health care [33]. Additionally, difficulty related to care coordination may also be related to the limitation of medium-complexity services at the beginning of the pandemic, caused by the reduction of health care professionals and increased focus on COVID-19, similar to what happened in other countries [34–37]. These factors also compromised care comprehensiveness, which obtained one of the worst scores.

The pandemic limited the offer of routine care in the PHC, impairing its performance in several countries [34, 38, 39]. In Brazil, in 2020, the Ministry of Health published the protocol for the "clinical management of coronavirus in primary health care," guiding the reorganization of PHC throughout the national territory [41]. However, many municipalities had difficulties managing and monitoring COVID-19 cases and responding to pandemic demands along with other PHC activities, evidencing the health care vulnerability and divergences among regions [40]. The demand of the pandemic competed with the regular health care demand, which is already challenging in Brazil [41].

Regarding the derived attributes, "community orientation" received a score lower than the cutoff point, corroborating surveys conducted in other Brazilian municipalities before the pandemic [15, 42, 43]. Community orientation exists as health also compasses the social context of individuals [7]. A European study identified the highest score of this attribute in rural communities and with more people from ethnic minorities [44].

BHU with partnerships with the "*Mais Médicos*" program or Family Medicine or Primary Care Residency did not impact our data significantly. However, the latter obtained a higher frequency of high overall scores. Although pre-pandemic studies indicated improvement in the quality of services in FHTs with a partnership with the "*Mais Médicos*" program [45, 46] or Family Medicine or Primary Care Residency [47], our study showed no differences between BHU with or without these programs during the COVID-19 pandemic.

Lower educational level was the characteristic of users most associated with high overall scores. Other studies conducted before the pandemic found similar results [14, 48], i.e., users who had less than secondary education gave slightly higher overall scores to health services than those with high school education or above. This result is probably because users with a higher educational level may be more critical when evaluating health services.

Regarding characteristics of BHU, the average employment time of professionals at the BHU was associated with high overall scores. The time of professionals in the same BHU helps the bond between them and the users, guaranteeing responsibility and regular attention and contributing to the longitudinality of care [7].

Several efforts were required to mitigate the effects of the COVID-19 pandemic on PHC worldwide, emphasizing providing remote care, controlling virus spread, monitoring patients diagnosed with COVID-19, seeking government support, and performing educational initiatives related to COVID-19 [21]. In Brazil, the pandemic coexisted with other PHC issues, triggered especially by changes in the National Primary Care Policy [49], which reduced the number of community health care professionals and flexibility in their workload [50]. Thus, PHC fragilities became more evident during the pandemic [48].

Thus, this study revealed insights into the quality and efficiency of health services offered during the COVID-19 pandemic. The attributes "first-contact care–accessibility" and bonding between professional and user showed positive results that may help strengthen health care.

This study also indicated barriers, such as poor accessibility and comprehensiveness of services, providing insights to improve the PHC organization. Moreover, the association between user satisfaction and educational level and longer employment time of professionals highlighted important aspects to elaborate public health policies. These findings may be a valuable guide for managers to develop effective and contextualized strategies that meet the needs of the population in different epidemiological scenarios.

Therefore, this study may help health management, indicating gaps for health professionals, suggesting strategies, and reinforcing the need to teach coping strategies during crises and skills and competencies for PHC work in pedagogical projects of Higher Education institutions.

The demands of users may change according to epidemiological conditions (e.g., the pandemic), justifying the relevance of our study. Information about health services helps managers identify weaknesses in the PHC, evidenced during the COVID-19 pandemic, strengthening health care. This study, however, has limitations. The use of a convenience sample may have led to a courtesy bias along with the inherent limitations of a cross-sectional design.

## Conclusion

The essential attributes and overall scores that users gave during the pandemic revealed that health services needed better orientation toward PHC. The attribute "First contact access–use" was the best evaluated, approaching the maximum score. Contrarily, "First contact access—accessibility" and "comprehensiveness" received the lowest average scores, suggesting that, although PHC is a reference to users, it did not fulfill their health needs. A significant association was found between lower educational level of users and high overall scores, indicating

these users are more satisfied with health services. The average employment time of professionals at the BHU was also associated with high overall scores, indicating the need for a bond between professionals and users to guarantee the quality of health services.

## Acknowledgments

The authors thank Probatus Academic Services for providing scientific language revision.

## Author Contributions

**Conceptualization:** Suely Deysny de Matos Celino, Nailton José Brandão de Albuquerque Filho, Monalisa da Nóbrega Cesarino Gomes, Gabriela Maria Cavalcanti Costa, Ana Elza Oliveira de Mendonça.

**Formal analysis:** Suely Deysny de Matos Celino, Nailton José Brandão de Albuquerque Filho, Monalisa da Nóbrega Cesarino Gomes, Gabriela Maria Cavalcanti Costa.

**Methodology:** Suely Deysny de Matos Celino, Nailton José Brandão de Albuquerque Filho, Monalisa da Nóbrega Cesarino Gomes, Gabriela Maria Cavalcanti Costa.

**Project administration:** Ana Elza Oliveira de Mendonça.

**Writing – original draft:** Suely Deysny de Matos Celino, Nailton José Brandão de Albuquerque Filho, Monalisa da Nóbrega Cesarino Gomes, Gabriela Maria Cavalcanti Costa.

**Writing – review & editing:** Suely Deysny de Matos Celino, Nailton José Brandão de Albuquerque Filho, Monalisa da Nóbrega Cesarino Gomes, Gabriela Maria Cavalcanti Costa, Ana Elza Oliveira de Mendonça.

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
