## [Decision Letter · Decision Letter 0]

4 Jul 2023

PONE-D-23-02410Evaluation of primary health care by users during the COVID-19 pandemicPLOS ONE

Dear Dr. Celino,

Thank you for submitting your manuscript to PLOS ONE. After careful consideration, we feel that it has merit but does not fully meet PLOS ONE’s publication criteria as it currently stands. Therefore, we invite you to submit a revised version of the manuscript that addresses the points raised during the review process.

We look forward to receiving your revised manuscript.

Kind regards,

Manoelito Ferreira Silva Junior, Ph.D.

Academic Editor

PLOS ONE

Journal Requirements:

3. Thank you for stating the following in your Competing Interests section: "The authors declare no conflicts of interest"

**Additional Editor Comments:**

I thank the authors for submitting the manuscript.

According to the reviewers' comments, I believe that the article needs a thorough review for the requested items, to qualify the methodology and understanding of the article.

Reviewers' comments:

Reviewer's Responses to Questions

**Comments to the Author**

1. Is the manuscript technically sound, and do the data support the conclusions?

Reviewer #1: No

Reviewer #2: Yes

2. Has the statistical analysis been performed appropriately and rigorously? 

Reviewer #1: No

Reviewer #2: Yes

3. Have the authors made all data underlying the findings in their manuscript fully available?

Reviewer #1: Yes

Reviewer #2: Yes

4. Is the manuscript presented in an intelligible fashion and written in standard English?

Reviewer #1: No

Reviewer #2: Yes

5. Review Comments to the Author

Reviewer #1: 1. Is the manuscript technically sound and do the data support the conclusions?

It is necessary to contextualize the study location more, for example: FHS coverage, number of health equipment in the PHC (UBS), number of professionals...

What are the exclusion criteria adopted?

Who is responsible for collections? Was there training? This information needs to be made clear.

It is necessary to explain the variables: “Advanced Access”, “More Doctors”, “Saúde na Hora”, “Residency in Family Medicine”. These are regional particularities that need to be explained in order to understand the theoretical logic that can justify its option for the bivariate study.

Why is the variable “COVID testing (0.227)” in table 4 of prevalence ratios PR, if the p value was greater than 0.2?

Regarding the PCATool, it is important to clarify the methodology for calculating the scores in relation to the answers “I don't know/I don't remember”: Missing value, median value or imputation. Ref.: Validating the Adult Primary Care Assessment Tool. The Journal of Family Practice. VOL. 50, NO. 2n 161W. FEBRUARY 2001.

Table 3 with the results of “High and Low Overall Score”, were the reference values >6.6 and <6.6 respectively adopted or the median, based on the value obtained (5.78 – DV 0.768)? It needs to be clear and since the categorization is by the median, it does not make much sense since the cutoff point is below the 6.6 points used as a reference in the “Methodology”.

2. Was the statistical analysis performed adequately and rigorously?

Even the variable “COVID testing" with p=0.227 shown in table 3”, is in table 4 of Prevalence Ratios RP. Would only variables with p value less than 0.2 be included in the methodology? This situation puts the statistical analysis in jeopardy doubt, because these are adjusted models and there may be influence on this adjustment because of a variable inserted in disagreement with the methodology.

3. Did the authors make available all the data underlying the findings in their manuscripts?

The authors undertake to make the data available.

4. Is the manuscript presented intelligibly and written in standard English?

Starting the first paragraph of the topic with descriptive prevalence data on chronic diseases and PHC is not a good way to go. Usually the main findings come first.

The results can be further explored, in why the attribute access - use was the best evaluated and this has the objective of identifying whether the PHC service is the reference for the user when he needs care and the worst evaluated, which is the access – accessibility that aims to identify whether the user somehow managed to be met in their need, since they have the service as a reference.

Reviewer #2: Thank you for the opportunity to read the manuscript "Evaluation of primary health care by users during the COVID-19 pandemic." The study is interesting and well conducted. However, there are aspects that need improvement. Below are some suggestions:

Introduction:

1.Title: I suggest including the study design.

2.I suggest explaining what the COVID-19 pandemic was and its epidemiological impacts on the Brazilian population.

3.In the sentence - "COVID-19 safety measures hindered PHC, but adaptations were strategically implemented, ensuring care continuity, planning, and organization to fulfill community needs" - I suggest providing a better description of the modifications made in PHC due to COVID-19.

4.I suggest explaining in the introduction what the PCA-TOOL is and how the assessment of PHC was before the pandemic.

5.The introduction does not reflect the study's intention. In addition to evaluating the essential attributes of PHC, there was a description of service delivery units, including the identification of programs such as "Mais Médicos," "Saúde na Hora"... Residência da Família... What are these programs? Why were they questioned? Is it because they could bring user satisfaction? What does the literature mention about this?

6.The study hypothesis needs to be clear.

7.In the sentence - "Therefore, this study aimed to evaluate PHC attributes and associated factors during the COVID-19 pandemic using the perspective of users" - it needs to clarify what these associated factors are and why they were chosen.

Materials and Methods:

8. What reference was used to describe the number of users in the PHC of Campina Grande?

9.Why was being vaccinated against COVID-19 an inclusion criterion?

10.Was there training for the field researchers? Was the questionnaire constructed by the authors pre-validated by a pilot study?

11.It is necessary to state that all participants signed the informed consent form.

12.In the sentence: "For the quantitative assessment of PHC, the expanded version for adults and the Brazilian population of the validated instrument Primary Care Assessment Tool (PCATool) was used," it is necessary to include the citation of the study that validates the instrument in Brazil.

Discussion:

13. I suggest starting the discussion by addressing the study hypothesis.

14.The attribute "first-contact care – use" obtained the best score, highlighting that users recognize the BHU as the first health care option. However, the attribute "first-contact care – accessibility" obtained the worst score. I suggest explaining these terms to the reader. The justifications for these findings are not clear. Why did first contact score well and accessibility score poorly? Did the municipality not offer telemedicine, leading to the low score? This statement needs to be supported by studies.

15.The average scores for "longitudinality" and "coordination - integrated care" attributes were slightly below the cutoff point. I suggest explaining these terms to the reader, as well as the reasons for these findings.

16. In light of the results, what do the authors suggest to the municipality? It is necessary to provide recommendations that contribute to the improvement of this service and for future studies.

6. PLOS authors have the option to publish the peer review history of their article (what does this mean?). If published, this will include your full peer review and any attached files.

Reviewer #1: No

Reviewer #2: No

---

## [Author Response · Author response to Decision Letter 0]

19 Aug 2023

Dr. Manoelito Ferreira Silva Júnior 

Academic Editor

PLOS ONE

Dear Dr. Manoelito Ferreira Silva Júnior,

We are resubmitting the paper entitled “Evaluation of primary health care by users during the COVID-19 pandemic" (PONE-D-23-02410). We have heeded the reviewer’s suggestions (red text in manuscript) and have addressed each issue below. The suggestions certainly helped improve the quality of the manuscript and we are most grateful for the feedback. 

REVIEWER #1

Reviewer´s comment: It is necessary to contextualize the study location more, for example: FHS coverage, number of health equipment in the PHC (UBS), number of professionals...

Authors’ response: We appreciate the comment. Was added to methodology: Data collection began in 2021 when the municipality had 110 FHT (comprising a medical doctor, a nurse, a nursing technician, and community health agents) distributed in 86 basic health units (BHU) and seven health districts, covering 91.78% of the Family Health Strategy [18]. 

 Despite the wide territorial coverage, formal guidelines, contingency plans, or municipal decrees of measures for PHC were not established in Campina Grande. The municipality restructured the hospital network, establishing an emergency care unit as a reference for COVID-19 screening and the municipal hospital for admissions, guiding PHC teams to refer for those services [19]. 

18. Municipal Secretary of Health (Campina Grande). Flow of regulation of beds 2nd health macro-region of Paraíba of suspected, probable and confirmed cases of the new coronavirus (COVID-19). Campina Grande, PB: MSH, 2020.

19. Harzheim E, Starfield B, Rajmil L, Álvarez-Dardet C, Stein AT. Internal consistency and reliability of Primary Care Assessment Tool (PCATool-Brasil) for child health services. Cad Saúde Pública. 2006; 22(8):1649–59.

Reviewer´s comment: What are the exclusion criteria adopted?

Authors’ response: Was added to methodology: “Those with cognitive difficulties in answering the questionnaire.”

Reviewer´s comment: Who is responsible for collections? Was there training? This information needs to be made clear.

Authors’ response: Was added to methodology: “Data collection was conducted by a single researcher. A pilot study with 20 users from two BHU was conducted to verify the data collection methodology and dynamics; changes were unnecessary. Users from the pilot study were not included in the main study. This snippet was added to the methodology.

Reviewer´s comment: It is necessary to explain the variables: “Advanced Access”, “More Doctors”, “Saúde na Hora”, “Residency in Family Medicine”. These are regional particularities that need to be explained in order to understand the theoretical logic that can justify its option for the bivariate study.

Authors’ response: Was added to introduction: “The federal government has been seeking strategies to strengthen PHC in Brazil, such as the "Mais Médicos" program and the promotion of family medicine and primary care residencies. These approaches help the training, provision, and settlement of medical doctors in remote areas while ensuring the effectiveness of care using incentives for professional training [10]. 

In 2019, the Ministry of Health instituted the "Saúde na Hora" program [11], which implemented extended hours in health units to increase access to health services. Additionally, Murray and Berwik [12] proposed Advanced Access, a strategy to improve user satisfaction within PHC. The slogan was 'Do today the work of today!', allowing users to meet their health needs, preferably on the same day, avoiding fragmentation of care [13].”

11. Ministério da Saúde (Brasil). Portaria nº 930, de 15 de maio de 2019. Institui o Programa "Saúde na Hora", que dispõe sobre o horário estendido de funcionamento das Unidades de Saúde da Família. [Internet] In: Diário Oficial da União, 2019. Available from: https://bvsms.saude.gov.br/bvs/saudelegis/gm/2019/prt0930_17_05_2019.html.

12. Murray M, Berwick DM. Advanced Access: Reducing Waiting and Delays in Primary Care

JAMA. 2003;289(8):1035-1040. doi:10.1001/jama.289.8.1035

13. Pires Filho LAS et al. Acesso Avançado em uma Unidade de Saúde da Família do interior do estado de São Paulo: um relato de experiência. Saúde Debate. 2019; 43(121): 605-13. doi: 10.1590/0103-1104201912124

Reviewer´s comment: Why is the variable “COVID testing (0.227)” in table 4 of prevalence ratios PR, if the p value was greater than 0.2?

Authors’ response: We apologize for the lack of clarity. The methodology employed was adopted using the rounding criterion for a decimal place. That way, the variable "teste COVID" presents p=0,20. was added to the methodology snippet “All variables with a p < 0.20 in the crude PR analysis were included in the initial adjusted model, having been rounded to one decimal place"

Reviewer´s comment: Regarding the PCATool, it is important to clarify the methodology for calculating the scores in relation to the answers “I don't know/I don't remember”: Missing value, median value or imputation. Ref.: Validating the Adult Primary Care Assessment Tool. The Journal of Family Practice. VOL. 50, NO. 2n 161W. FEBRUARY 2001.

Authors’ response: Was added to methodology: According to the Brazilian PCATool Manual [20], the percentage of items with missing values (blank values or answers with code "9" - "don't know, don't remember") should be under 50% of the total number of attribute items to calculate user score.

20. Ministério da Saúde (Brasil). Manual of the Primary Health Care Assessment Instrument: PCATool. Brasília: Ministério da Saúde; 2020.

Reviewer´s comment: Table 3 with the results of “High and Low Overall Score”, were the reference values >6.6 and <6.6 respectively adopted or the median, based on the value obtained (5.78 – DV 0.768)? It needs to be clear and since the categorization is by the median, it does not make much sense since the cutoff point is below the 6.6 points used as a reference in the “Methodology”.

Authors’ response: Was added: Table 3 shows the high (score ≥ 6.6) and low (score < 6,6) overall score distribution according to the characteristics of users and BHU.

Reviewer´s comment: Even the variable “COVID testing" with p=0.227 shown in table 3”, is in table 4 of Prevalence Ratios RP. Would only variables with p value less than 0.2 be included in the methodology? This situation puts the statistical analysis in jeopardy doubt, because these are adjusted models and there may be influence on this adjustment because of a variable inserted in disagreement with the methodology.

Authors’ response: We apologize for the lack of clarity. The methodology employed was adopted using the rounding criterion for a decimal place. That way, the variable "teste COVID" presents p=0,20. was added to the methodology snippet “All variables with a p < 0.20 in the crude PR analysis were included in the initial adjusted model, having been rounded to one decimal place"

Reviewer´s comment: Starting the first paragraph of the topic with descriptive prevalence data on chronic diseases and PHC is not a good way to go. Usually the main findings come first.

Authors’ response: We appreciate the comment. Was modified the first paragraph of discussion: “This study demonstrated that higher levels of education of users and longer employment time of professionals were associated with better PHC attributes. The municipality scored under the minimum score 6.6 for essential attributes, indicating services without the minimum PHC characteristics.”

Reviewer´s comment: The results can be further explored, in why the attribute access - use was the best evaluated and this has the objective of identifying whether the PHC service is the reference for the user when he needs care and the worst evaluated, which is the access – accessibility that aims to identify whether the user somehow managed to be met in their need, since they have the service as a reference.

Authors’ response: We appreciate the comment. Was added to discussion: In Brazil, poor accessibility was present before the pandemic and reflected organizational issues (e.g., reduced opening hours, difficulty scheduling appointments, and long waits for care) [25]. 

A study conducted with PHC health professionals and managers of Brazilian municipal health departments observed actions on health surveillance and comprehensive care. The actions were performed via telephone, WhatsApp app, and home visits of community health agents to guarantee access and pandemic control by PHC. Moreover, the study indicated a barrier: need for more supply, internet, personal protective equipment, reverse transcription polymerase chain reaction (RT-PCR) tests, and continuing education for professionals [26].

25. Challenges of Primary Care in Confronting the Covid-19 Pandemic in the SUS. In: Portela MC, Reis LGC, Lima, SML (org). Covid-19: Challenges for the organization and repercussions on health systems and services. Rio de Janeiro: Fiocruz; 2022: 201-16.

26. Oliveira-Morais J, Morais F, Santiago C. First contact access in primary health care for children from 0 to 9 years old. Rev Pesqui Cuid Fundam. 2017;9(3):848-56. doi: 10.9789/2175-5361.2017.v9i3.848-856.

REVIEWER #2

Introduction:

Reviewer´s comment: Title: I suggest including the study design.

Authors’ response: We appreciate the comment. The type of study was added to the title.

“Evaluation of primary health care by users during the COVID-19 pandemic: a cross-sectional study”

Reviewer´s comment: I suggest explaining what the COVID-19 pandemic was and its epidemiological impacts on the Brazilian population.

Authors’ response: We appeciate the suggestion. Was added to introduction: “In December 2019, Wuhan, the capital of Hubei, China, reported the first coronavirus disease 2019 (COVID-19) cases. The disease presented quick spread and severity, reaching other province regions and worldwide, characterizing a pandemic after China, Germany, Japan, Vietnam, and the United States registered human-to-human transmission [1]. Then, in March 2020, the World Health Organization (WHO) declared the global outbreak of COVID-19 as an international emergency and recommended the implementation of consistent and evidence-based decisions [2].

Although differently in each country, the COVID-19 impacted health services worldwide, since they were unprepared for a pandemic [3]. […] The PHC, by being first contact of the population with the health system, was essential during the COVID-19 pandemic and its reorganization was specially challenging [6].”

1. Kang D et al. Spatial epidemic dynamics of the COVID-19 outbreak in China. Int J Infect Dis. 2020; 94: 96-102.

2. WHO (World Health Organization). Pan American Health Organization. WHO states that COVID-19 is now characterized as a pandemic. 2020. Available from: https://www.paho.org/bra/index.php?option=com_content&view=article&id=6120:oms-afirma-que-covid-19-e-agora-caracterizada-como-pandemia&Itemid=812

3. Mustafa S. COVID-19 Preparedness and Response Plans from 106 countries: a review from a health systems resilience perspective. Health Policy Plan. 2022; 37(2): 255-68.

6. Subba SH; Pradhan SK; Sahoo BK. Empowering primary healthcare institutions against COVID-19 pandemic: a health system-based approach. J Family Med Prim Care. 2021; 10(2):589-94.

Reviewer´s comment: In the sentence - "COVID-19 safety measures hindered PHC, but adaptations were strategically implemented, ensuring care continuity, planning, and organization to fulfill community needs" - I suggest providing a better description of the modifications made in PHC due to COVID-19.

Authors’ response: We appreciate the comment. Was added to introduction: In Brazil, the Ministry of Health published ordinances, manuals, guides, and protocols with information on symptoms, transmission forms, clinical management, and guidelines to prevent COVID-19 infection [4]. In April 2020, the “protocol for clinical management of coronavirus (COVID-19) in primary health care (PHC)” was published and aimed to "define the role of PHC services in the management and control of COVID-19 infection, and to provide clinical guidance tools for professionals" but without instructions for restructuring actions of PHC [5].

4. Fernandez M, Fernandes L da MM, Massuda A. Primary Health Care in the COVID-19 pandemic: an analysis of response plans to the health crisis in Brazil . Rev Bras Med Fam Comunidade. 2022;17(44):3336. 

5. Ministério da Saúde (Brasil). Clinical Management Protocol for the Coronavirus (Covid-19) in Primary Health Care. Brasília: Ministério da Saúde, 2020.

Reviewer´s comment: I suggest explaining in the introduction what the PCA-TOOL is and how the assessment of PHC was before the pandemic.

Authors’ response: We appeciate the suggestion. Was added to introduction: “Thus, assessing the pandemic impact on the quality of care and work process of PHC teams is important to improve action plans and adaptation. According to Starfield [7], a PHC assessment requires identifying whether PHC attributes guided the health services, improving health indicators, user satisfaction, costs, equity, and the health status of the population.

The Primary Care Assessment Tool (PCATool) is a structured questionnaire developed to assess PHC essential (first-contact care, longitudinality, comprehensiveness, and coordination) and derived (family and community orientation and cultural competence) attributes using the perspective of users and health professionals. The instrument is in the public domain, used by WHO, and was validated in different countries (e.g., Brazil, South Korea, and Spain) [8].” 

7. Starfield B. Primary care: balancing health needs, services and technology. Brasilia: UNESCO, Ministry of Health; 2002.

8. Prates ML, Machado JC, Silva LS da, Avelar PS, Prates LL, Mendonça ET de, et al.. Performance of primary health care according to PCATool instrument: a systematic review. Ciênc Saúde Coletiva. 2017; 22(6):1881–93.

Reviewer´s comment: The introduction does not reflect the study's intention. In addition to evaluating the essential attributes of PHC, there was a description of service delivery units, including the identification of programs such as "Mais Médicos," "Saúde na Hora"... Residência da Família... What are these programs? Why were they questioned? Is it because they could bring user satisfaction? What does the literature mention about this?

Authors’ response: We appreciate the comment. Was added to introduction: “The perception of the quality of health services is determined by sociodemographic characteristics of users and inherent factors [9]. The federal government has been seeking strategies to strengthen PHC in Brazil, such as the "Mais Médicos" program and the promotion of family medicine and primary care residencies. These approaches help the training, provision, and settlement of medical doctors in remote areas while ensuring the effectiveness of care using incentives for professional training [10]. 

In 2019, the Ministry of Health instituted the "Saúde na Hora" program [11], which implemented extended hours in health units to increase access to health services. Additionally, Murray and Berwik [12] proposed Advanced Access, a strategy to improve user satisfaction within PHC. The slogan was 'Do today the work of today!', allowing users to meet their health needs, preferably on the same day, avoiding fragmentation of care [13].”

9. Augusto DK, Lima-Costa MF, Macinko J, Peixoto SV. Factors associated with the evaluation of quality of primary health care by older adults living in the Metropolitan Region of Belo Horizonte, Minas Gerais, Brazil, 2010. Epidemiol Serv Saúde; 2019; 28(1):e2018128. 

10. Barrêto D da S, Melo Neto AJ de, Figueiredo AM de, Sampaio J, Gomes LB, Soares R de S. The More Doctors Program and Family and Community Medicine residencies: articulated strategies of expansion and interiorization of medical education. Interface; 2019; 23:e180032. 

11.Ministério da Saúde (Brasil). Portaria nº 930, de 15 de maio de 2019. Institui o Programa "Saúde na Hora", que dispõe sobre o horário estendido de funcionamento das Unidades de Saúde da Família. [Internet] In: Diário Oficial da União, 2019. Available from: https://bvsms.saude.gov.br/bvs/saudelegis/gm/2019/prt0930_17_05_2019.html.

12. Murray M, Berwick DM. Advanced Access: Reducing Waiting and Delays in Primary Care. JAMA. 2003;289(8):1035-1040. doi:10.1001/jama.289.8.1035

13. Pires Filho LAS et al. Acesso Avançado em uma Unidade de Saúde da Família do interior do estado de São Paulo: um relato de experiência. Saúde Debate. 2019; 43(121): 605-13. doi: 10.1590/0103-1104201912124

Reviewer´s comment: The study hypothesis needs to be clear.

Authors’ response: Was added to introduction: “Thus, we hypothesized that worse sociodemographic factors and PHC conditions would be associated with unmet attributes.”

Reviewer´s comment: "Therefore, this study aimed to evaluate PHC attributes and associated factors during the COVID-19 pandemic using the perspective of users" - it needs to clarify what these associated factors are and why they were chosen.

Authors’ response: Was added to introduction: “associating the satisfaction of users with sociodemographic and service characteristics may reveal important aspects of public health policies. […] The study aimed to assess PHC during the COVID-19 pandemic, considering its attributes and using the perspective of users and associated factors."

Materials and Methods:

Reviewer´s comment: What reference was used to describe the number of users in the PHC of Campina Grande?

Authors’ response: e-SUS: National Primary Care Information System.

Ministério da Saúde (Brasil). Informação e Gestão da Atenção Básica: e-Gestor AB. 2021. Available from: https://egestorab.saude.gov.br/.

Reviewer´s comment: Why was being vaccinated against COVID-19 an inclusion criterion?

Authors’ response: Due to the high contamination, the municipal health department demanded the vaccination of the researcher and users for in-person data collection. This snippet was added to the methodology.

Reviewer´s comment: Was there training for the field researchers? Was the questionnaire constructed by the authors pre-validated by a pilot study?

Authors’ response: Data collection was conducted by a single researcher. A pilot study with 20 users from two BHU was conducted to verify the data collection methodology and dynamics; changes were unnecessary. Users from the pilot study were not included in the main study. This snippet was added to the methodology.

Reviewer´s comment: It is necessary to state that all participants signed the informed consent form.

Authors’ response: Was added: “Users were informed of the study aims and signed the informed consent form; confidentiality and anonymity were guaranteed.”

Reviewer´s comment: In the sentence: "For the quantitative assessment of PHC, the expanded version for adults and the Brazilian population of the validated instrument Primary Care Assessment Tool (PCATool) was used," it is necessary to include the citation of the study that validates the instrument in Brazil.

Authors’ response: We appreciate the comment. The reference has been included.

Discussion:

Reviewer´s comment: I suggest starting the discussion by addressing the study hypothesis. 

Authors’ response: We appreciate the comment. Was modified the first paragraph of discussion: “This study demonstrated that higher levels of education of users and longer employment time of professionals were associated with better PHC attributes. The municipality scored under the minimum score 6.6 for essential attributes, indicating services without the minimum PHC characteristics.”

Reviewer´s comment: The attribute "first-contact care – use" obtained the best score, highlighting that users recognize the BHU as the first health care option. However, the attribute "first-contact care – accessibility" obtained the worst score. I suggest explaining these terms to the reader. The justifications for these findings are not clear. Why did first contact score well and accessibility score poorly? Did the municipality not offer telemedicine, leading to the low score? This statement needs to be supported by studies.

Authors’ response: We appreciate the comment. Was added: “In Brazil, poor accessibility was present before the pandemic and reflected organizational issues (e.g., reduced opening hours, difficulty scheduling appointments, and long waits for care) [25]. 

A study conducted with PHC health professionals and managers of Brazilian municipal health departments observed actions on health surveillance and comprehensive care. The actions were performed via telephone, WhatsApp app, and home visits of community health agents to guarantee access and pandemic control by PHC. Moreover, the study indicated a barrier: need for more supply, internet, personal protective equipment, reverse transcription polymerase chain reaction (RT-PCR) tests, and continuing education for professionals [26].”

25. Challenges of Primary Care in Confronting the Covid-19 Pandemic in the SUS. In: Portela MC, Reis LGC, Lima, SML (org). Covid-19: Challenges for the organization and repercussions on health systems and services. Rio de Janeiro: Fiocruz; 2022: 201-16.

26. Oliveira-Morais J, Morais F, Santiago C. First contact access in primary health care for children from 0 to 9 years old. Rev Pesqui Cuid Fundam. 2017;9(3):848-56. doi: 10.9789/2175-5361.2017.v9i3.848-856.

Reviewer´s comment: The average scores for "longitudinality" and "coordination - integrated care" attributes were slightly below the cutoff point. I suggest explaining these terms to the reader, as well as the reasons for these findings.

Authors’ response: We appreciate the comment. Was added: “Most professionals had over a year of employment time, favoring bonding, which is fundamental to the longitudinality of care. However, poor accessibility to services and other attributes may hinder operationalization [30]. 

The attribute "coordination" improves the PHC quality by enhancing the accessibility to other levels of care, integrating actions, services, and professionals vertically and horizontally, and linking community and individual resources [31]. Poor care coordination hinders managing non-communicable and infectious diseases [32], especially during the COVID-19 pandemic, when PHC is fundamental in organizing the care network and guaranteeing an efficient flow [33].”

30. Li, X et al. Quality of primary health care in China: challenges and recommendations. Lancet. 2020; 6 (12) :1802–12. doi: 10.1016/S0140-6736(20)30122-7

31. Ximenes Neto FRG et al. Coordenação do cuidado, vigilância e monitoramento de casos da covid-19 na atenção primária à saúde. Enferm. Foco 2020; 11 (1) Especial: 239-245.

32. Ministério da Saúde (Brasil). Protocolo de Manejo Clínico do Coronavírus (Covid-19) na Atenção Primária à Saúde. Brasília: Ministério da Saúde; 2020.

33. Plagg B, Piccoliori G, Oschmann J, Engl A, Eisendle K. Primary Health Care and Hospital Management During COVID-19: Lessons from Lombardy. Risk Manag Healthc Policy. 2021;14:3987-92. doi: 10.2147/rmhp.s315880. PubMed PMID: 34602826.

Reviewer´s comment: In light of the results, what do the authors suggest to the municipality? It is necessary to provide recommendations that contribute to the improvement of this service and for future studies.

Authors’ response: We appreciate the comment. Was added: “Thus, this study revealed insights into the quality and efficiency of health services offered during the COVID-19 pandemic. The attributes “first-contact care – accessibility” and bonding between professional and user showed positive results that may help strengthen health care. 

This study also indicated barriers, such as poor accessibility and comprehensiveness of services, providing insights to improve the PHC organization. Moreover, the association between user satisfaction and educational level and longer employment time of professionals highlighted important aspects to elaborate public health policies. These findings may be a valuable guide for managers to develop effective and contextualized strategies that meet the needs of the population in different epidemiological scenarios. 

Therefore, this study may help health management, indicating gaps for health professionals, suggesting strategies, and reinforcing the need to teach coping strategies during crises and skills and competencies for PHC work in pedagogical projects of Higher Education institutions.”

We thank the reviewers for their contributions.

The authors

---

## [Decision Letter · Decision Letter 1]

12 Sep 2023

Evaluation of primary health care by users during the COVID-19 pandemic: a cross-sectional study

PONE-D-23-02410R1

Dear Dra. Celino

We’re pleased to inform you that your manuscript has been judged scientifically suitable for publication and will be formally accepted for publication once it meets all outstanding technical requirements.

Kind regards,

Manoelito Ferreira Silva Junior, Ph.D.

Academic Editor

PLOS ONE

I appreciate the authors' efforts to address the reviewers' points. After the considerations presented by the authors, the article presents all the points to be published in the current version.

Reviewers' comments:

Reviewer's Responses to Questions

**Comments to the Author**

1. If the authors have adequately addressed your comments raised in a previous round of review and you feel that this manuscript is now acceptable for publication, you may indicate that here to bypass the “Comments to the Author” section, enter your conflict of interest statement in the “Confidential to Editor” section, and submit your "Accept" recommendation.

Reviewer #2: All comments have been addressed

2. Is the manuscript technically sound, and do the data support the conclusions?

Reviewer #2: Yes

3. Has the statistical analysis been performed appropriately and rigorously? 

Reviewer #2: Yes

4. Have the authors made all data underlying the findings in their manuscript fully available?

Reviewer #2: Yes

5. Is the manuscript presented in an intelligible fashion and written in standard English?

Reviewer #2: No

6. Review Comments to the Author

Reviewer #2: (No Response)

7. PLOS authors have the option to publish the peer review history of their article (what does this mean?). If published, this will include your full peer review and any attached files.

Reviewer #2: No

---

## [Editor Report · Acceptance letter]

15 Sep 2023

PONE-D-23-02410R1 

Evaluation of primary health care by users during the COVID-19 pandemic: a cross-sectional study 

Dear Dr. Celino:

I'm pleased to inform you that your manuscript has been deemed suitable for publication in PLOS ONE. Congratulations! Your manuscript is now with our production department. 

Kind regards, 

on behalf of

Dr. Manoelito Ferreira Silva Junior 

Academic Editor

PLOS ONE